# Reliability and validity of a low-cost, wireless sensor and smartphone app for measuring force during isometric and dynamic resistance exercises

Víctor Illera-Domínguez[1], Lluís Albesa-Albiol[1]*, Jorge Castizo-Olier[1,2‡], Adrián Garcia-Fresneda[1,3‡], Bernat Buscà[4‡], Carlos Ramirez-Lopez[5,6‡], Bruno Fernández-Valdés[1]

**1** Department of Health Sciences, Research Group in Technology Applied to High Performance and Health, TecnoCampus, Universitat Pompeu Fabra, Mataró, Barcelona, Spain, **2** INEFC-Barcelona Sports Sciences Research Group, Institut Nacional d'Educació Física de Catalunya (INEFC), Universitat de Barcelona, Barcelona, Spain, **3** Institut Nacional d'Educació Física de Catalunya (INEFC), Universitat de Barcelona, Barcelona, Spain, **4** Blanquerna - Universitat Ramon Llull, Barcelona, Spain, **5** Carnegie Applied Rugby Research (CARR) Centre, Institute for Sport, Physical Activity and Leisure, Carnegie School of Sport, Leeds Beckett University, Leeds, United Kingdom, **6** Scottish Rugby Union, Murrayfield Stadium, Edinburgh, United Kingdom

☯ These authors contributed equally to this work.
‡ JCO, AGF, BB and CRL also contributed equally to this work.
* lalbesa@tecnocampus.cat

## Abstract

The aim of this study was to investigate the reliability and validity of an affordable wireless force sensor in measuring mean and peak forces during resistance training. A Suiff Pro wireless force sensor (Suiff, Spain) and a MuscleLab force platform (Ergotest, Norway) were used concurrently to assess tensile load and the ground reaction force resulting from an upright row exercise. Thirteen participants (28.2 ± 5.7 years, 76.2 ± 9.6 kg, 178.2 ± 9.2 cm) performed the exercise under three velocity conditions and isometrically. Each condition involved three sets of exercise. Mean ($F_{mean}$) and peak ($F_{peak}$) force values from both sensors were collected and compared. Suiff Pro exhibited excellent reliability for $F_{mean}$ and $F_{peak}$ (ICCs = 0.99). When compared to the criterion measures, Suiff Pro showed trivial standardized bias for $F_{mean}$ (Mean = 0.00 [CI 95% = 0.00 to 0.01]) and $F_{peak}$ (-0.02 [-0.04 to 0.00]). The standardized typical error was also trivial for $F_{mean}$ (0.03 [0.02 to 0.03]) and $F_{peak}$ (0.07 [0.05 to 0.09]). Correlations with the MuscleLab force platform were nearly perfect: $F_{mean}$ (0.97 [0.94 to 0.98]; p<0.001); $F_{peak}$ (0.96 [0.92 to 0.97]; p<0.001). The findings demonstrate that the Suiff Pro sensor is reliable and valid device for measuring force during isometric and dynamic resistance training exercises. Therefore, practitioners can confidently use this device to monitor kinematic variables of resistance training exercises and to obtain real-time augmented feedback during a training session.

**Data Availability Statement:** All relevant data are within the paper and its Supporting Information files. We have submitted an Excel file with the data registry.

**Funding:** This study has received approval for support in publication fees from TecnoCampus - Universitat Pompeu Fabra - Research group in Technology Applied to High Performance and Health (TAARS) (Resolution 33/2023).

**Competing interests:** The company, Suiff, received funding for product development from the INEFC of Barcelona. Two of the authors, Jorge Castizo-Olier and Adrián Garcia-Fresneda, are affiliated with the INEFC of Barcelona as educators. This does not alter our adherence to PLOS ONE policies on sharing data and materials. There are no patents, products in development or marketed products associated with this research to declare.

## Introduction

Technological advancements in sensors and data management have benefited the fields of sports performance and physical activity for health by enabling continuous collection of vast amounts of data from individuals regarding their activity, rest, and performance during training and competitions [1]. This has allowed for a better understanding of the training process, as well as greater control and individualization of training and rehabilitation programs [2]. Monitoring an athlete's training load is important for adequate training periodisation and for adjusting training doses in order to avoid undesirable situations such as non-functional over-reaching [3]. Furthermore, live feedback of training loads and physical outputs has been shown to be useful for improving performance during a session, and for ultimately augmenting the effectiveness of the training process [4].

Resistance training (RT) constitutes a fundamental component in the domains of sports performance and health [5, 6]. However, quantifying and monitoring RT in a way that is valid, reliable, and practical may be a challenging task, given that a typical RT session may involve various methods (e.g., gravity-dependent loads, iso-inertial training, elastics, pneumatic machines, etc.) [7, 8]. Force outputs are one of the most important and commonly used parameters for live monitoring of RT (i.e., live augmented feedback) and to evaluate progress throughout a training cycle [9]. Dynamometers of varying properties (e.g., single or multi-component load cells), typically in the form of force platforms, isokinetic devices, hand-held dynamometers (i.e., manual-pressure) or force gauges, can be utilized to evaluate force outputs, with some of these systems being more appropriate than the others depending on the situation [1]. Force outputs can be assessed (I) alone (i.e., peak or mean forces during a given period), (II) in combination with temporal parameters (e.g., rate of force development), or (III) combined with spatial-temporal variables to compute power or work (e.g., when a force sensor is used in combination with a linear encoder during dynamic exercises) [9–12].

Force platforms are the gold standard method for measuring ground reaction forces during different exercises (e.g., jumps, squats, etc.) [13, 14]. While force platforms are highly precise, their main drawbacks are their cost, low portability, and their inability to assess force in movements where force is not applied against the ground such as upper body pulls. As such, alternative assessments like isokinetic dynamometry which allow the assessment of force and torque through the range of motion of a single joint have also been considered a gold standard and an alternative to force platforms [15]. However, the main drawbacks of these systems also include their cost, low portability, and inability to utilise with multi-joint exercises [16]. Hand-held dynamometers represent a versatile, portable and more affordable system for the assessment of force outputs and have therefore been widely used in clinical and public health settings [17]. However, the assessment can have a low reliability because of human factors such as anthropometric characteristics and strength of the examiner. Furthermore, hand-held dynamometers typically provide only a single metric of maximal isometric strength and no information about the force-time characteristics of the assessment [18]. Force gauges which consist of a tensile load sensor with two opposed handles have also been utilised as an alternative for the assessment of force outputs. The gauges can be connected to different RT systems with carabiners (e.g., cables, elastic bands, belts, suspension training anchors, iso-inertial devices), making them useful for monitoring a variety of exercises [19]. Until recently, these sensors were connected to a central processing unit through a wire, making them costly, less portable and usable than recently developed models. Although the practicality of low-cost force gauges are clear, new devices have to prove their criterion validity, in order to provide confidence in data quality to guide the training and rehabilitation programs [20–22].

The Suiff Pro load sensor (Suiff, Barcelona, Spain) is a low-cost wireless new generation force gauge which features the ability to provide live feedback during RT exercises [23]. The device uses a monoaxial tensile load sensor connected by Bluetooth to a smartphone or tablet to measure the force-time curve, peak and mean force developed in real time. Raw data can also be exported to compute other parameters, such as the rate of force development. However, the reliability and validity of new generation force gauges such as the Suiff Pro sensor has not been previously assessed. Therefore, the aim of this study was to assess the reliability and validity of the Suiff Pro wireless force sensor for the assessment of peak and mean forces during isometric and dynamic RT exercises. We hypothesized the device would demonstrate reliable measurements and a high level of agreement with force platforms.

## Materials and methods

### Experimental approach to the problem

This study aimed to assess the reliability and criterion validity of force values obtained using the Suiff Pro wireless force sensor (Suiff Pro, Suiff, Barcelona, Spain) by measuring several sets of upright rows (maximal sets under isometric conditions and submaximal sets resisted by elastic bands at various velocities) simultaneously with (I, practical measure) the Suiff Pro sensor and (II, criterion measure) a force platform. The degree of agreement between force data acquired from both methods was subsequently evaluated.

### Subjects

Thirteen (11 male, 2 female) physically active adult volunteers (age: 28.2 ± 5.7 years, body mass: 76.2 ± 9.6 kg, height: 178.2 ± 9.2 cm) with no injury were recruited to participate in this study. All subjects had prior experience with the exercise technique and testing protocols. The experimental procedures were approved by the ethics committee of the Catalan Sports Council (Government of Catalonia) No. 032/CEICGC/2022, and written consent was obtained from all subjects prior to study initiation. The recruitment phase for this investigation spanned from November 20, 2022, to January 15, 2023. The individual pictured in Fig 1 has provided written informed consent (as outlined in PLOS consent form) to publish their image alongside the manuscript.

### Procedures

To assess agreement between measuring devices in diverse loading conditions, participants performed upright rows under four different conditions: (A) 20 seconds of maximal isometric upright row resisted by a non-elastic strap at approximately 90˚ elbow angle; (B) dynamic submaximal sets of 17 repetitions of upright rows resisted by elastic bands at 60, (C) 90, and (D) 120 beats per minute (BPM). Full range of motion was required for each repetition during the dynamic conditions, with hands moving from the sternum to full extension of the elbows, matching initial and final position with the metronome beats (Soundbrenner App, Los Angeles, USA) at the specified speeds. Any set execution not meeting these parameters was discarded. The order in which participants performed under each of the four conditions was randomized, with three consecutive sets of exercise recorded for each condition. A 3-minute recovery period was provided between sets. Force was simultaneously assessed using both methods during each set of the exercise (Fig 1).

Criterion measures of force were obtained by assessing the resultant ground reaction forces using a MuscleLab 6000 triaxial force platform (Ergotest Technology, Porsgrunn, Norway), which has been previously utilized as a reference system for kinetic variables [12, 14]. Subjects

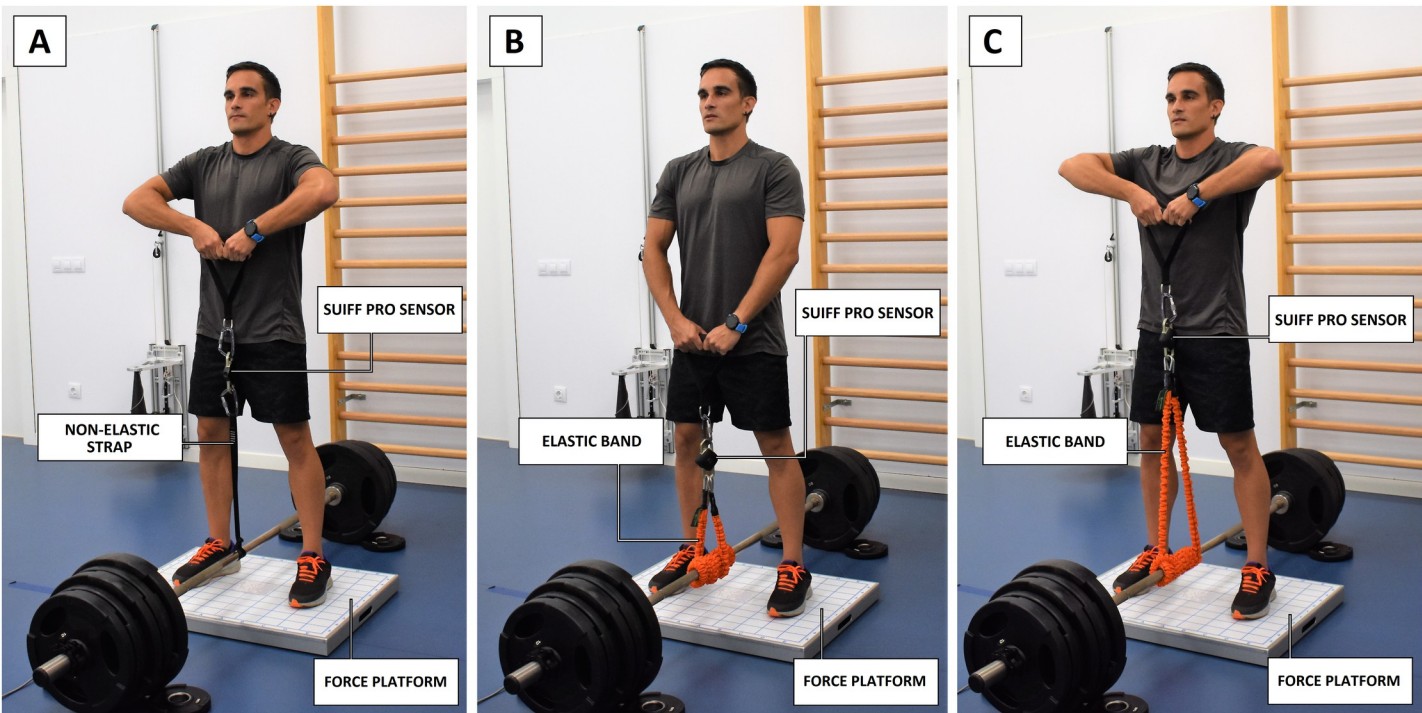

**Fig 1. Exercise protocols and disposition of the different sensors used in the study.** (A) Maximal isometric upright row resisted by a non-elastic strap. (B) Dynamic submaximal sets of upright rows resisted by elastic bands, initial position. (C) Dynamic submaximal sets of upright rows resisted by elastic bands, final position.

were positioned on the force platform (capacity: 2000 kg, sampling rate: 1000 Hz), and the resistance (strap or elastic band) was attached to a fixed barbell located at the center between the legs, without contact with the platform (Fig 1).

The Suiff Pro wireless monoaxial force sensor (Suiff, Barcelona, Spain) was attached between the handles and the strap or elastic bands, depending on the condition (Fig 1). The technical specifications and detailed view of the sensor provided by the manufacturer are shown in Fig 2.

In each set of exercise, measurements of the set mean force ($F_{mean}$) and set peak force ($F_{peak}$) were obtained from both sensors.

## Statistical analysis

The data were imported into JASP software version 0.17.3.0 for Windows (JASP, Amsterdam, The Netherlands). The normality of the data, presented as means and standard deviations (SD), was assessed using the Shapiro-Wilk test. To assess the reliability of the measurement devices concerning $F_{mean}$ and $F_{peak}$ a test-retest reliability analysis was conducted using the data from the three consecutive sets. Specifically, the intraclass correlation (ICC) 3,1 with 95% confidence intervals (CI) was applied according to the methodology proposed by Shrout and Fleiss [24]. Additionally, standardized typical error (TE) with 95% CI was calculated using a purpose-built Microsoft Excel spreadsheet for reliability analysis [25]. The level of agreement between criterion and practical measures of $F_{mean}$ and $F_{peak}$ from the first set was evaluated using correlation analysis. The Pearson correlation test was employed for datasets conforming to a normal distribution whereas the Spearman's rho test was utilized for datasets that did not meet the assumption of normality. The outcomes of both tests were presented alongside their respective 95% CI and statistical significance was considered at a significance level of $p < 0.05$.

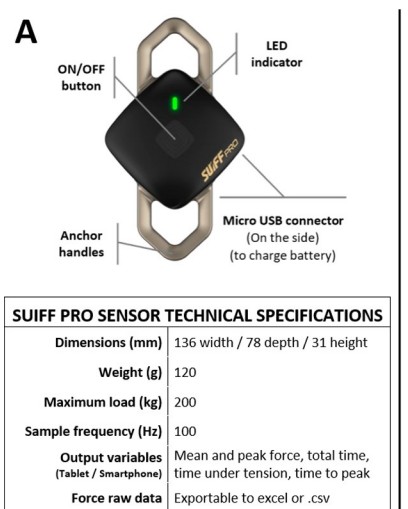
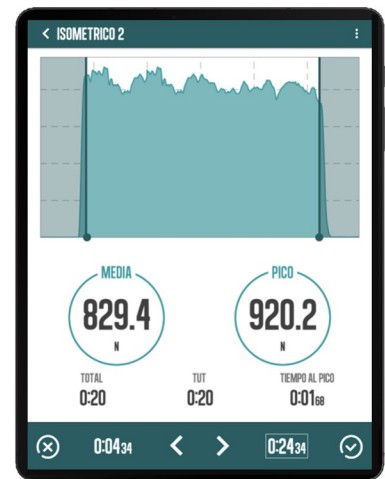

**Fig 2. Characteristics of the Suiff pro sensor (Suiff, Barcelona, Spain).** (A) Detailed view and technical specifications provided by the manufacturer. (B) Example of a recorded isometric upright row when accessed through the tablet interface, data visualization is provided in real time when recording. Reprinted from Suiff under a CC BY license, with permission from Suiff, original copyright 2021.

In addition, standardized TE and mean bias with 95% CI were calculated using a purpose-built Microsoft Excel spreadsheet for validity analysis [25]. Furthermore, Bland–Altman plots were employed to visually supplement the differences between the two systems [26]. As proposed by Hopkins [25], the magnitude of correlation was rated as trivial ($<0.1$), small ($0.1–0.29$), moderate ($0.3–0.49$), large ($0.5–0.69$), very large ($0.7–0.89$), or nearly perfect ($0.9–0.99$); the standardized TE was rated as trivial ($<0.1$), small ($0.1–0.29$), moderate ($0.3–0.59$), or large ($>0.59$); and the standardized mean bias was rated as trivial ($\leq0.19$), small ($0.2–0.59$), moderate ($0.6–1.19$), or large ($1.2–1.99$). Fig 3 was generated using GraphPad Prism 9.2 for Windows (GraphPad Software Inc., San Diego, USA).

## Results

Excellent reliability (ICC) was noted for both devices across the three sets for $F_{mean}$ and $F_{peak}$, indicating strong reproducibility (Table 1).

In comparison to the criterion measures, the practical measures of $F_{mean}$ and $F_{peak}$ showed trivial mean bias across all conditions. TE for both parameters was also trivial when all data were pooled, although $F_{peak}$ TE tended to increase with increasing execution pace. Correlations between the two methods were nearly perfect for $F_{mean}$ and $F_{peak}$ in all load configurations (Table 2).

The regression equations to estimate the criterion measures from the practical measures fitted nearly perfect with data (Fig 3A and 3C) and are presented as follows:

$$Y = \text{intercept} + (\text{slope} \cdot X)$$

where Y is the estimated criterion measure and X is the practical measure.

The regression equation of $F_{mean}$ is:

$$Y = 0.4181 + (0.9959 \cdot X)$$

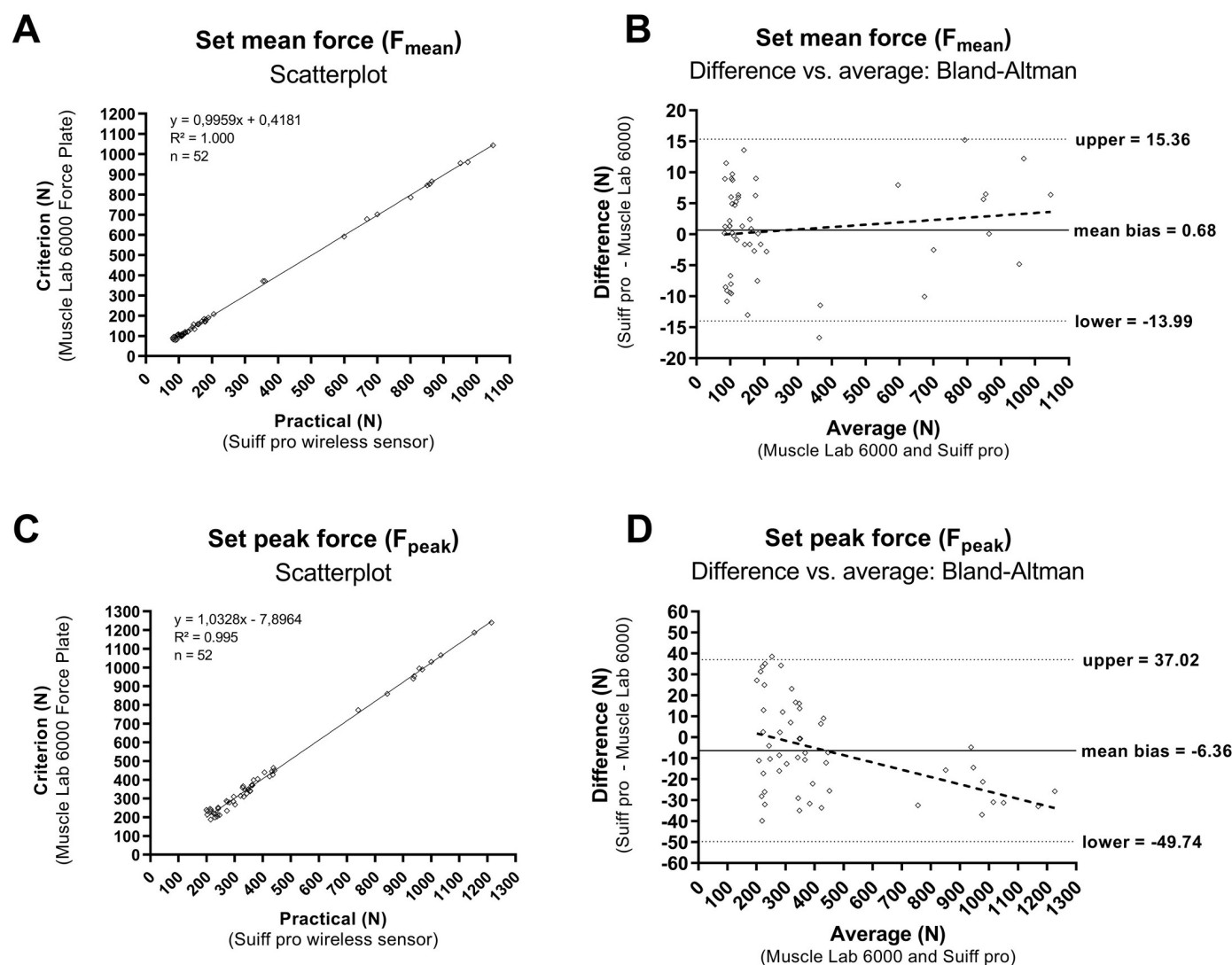

**Fig 3.** Scatterplots showing the agreement between the criterion and practical measures Fmean (A) and Fpeak (C). Bland-Altman plots for Fmean (A) and Fpeak (C). The solid horizontal line within the graphs represents the mean bias. The broken horizontal lines represent the upper and lower 95% limits of agreement. The regression lines of the scattered points are shown to investigate the homoscedasticity of the errors.

**Table 1. Intraclass correlations (ICC 3,1) for test–retest reliability of $F_{mean}$ and $F_{peak}$ measured with MuscleLab 6000 force platform and Suiff pro sensor in three consecutive sets of exercise.**

| Variable | Device | Set1 | Set 2 | Set 3 | TE | ICC (3,1) |
|---|---|---|---|---|---|---|
| $F_{mean}$ (n = 52) | Suiff sensor | 269.5 ± 288.5 N | 268.1 ± 289.4 N | 262.2 ± 265.9 N | 0.12 [0.11 to 0.15] (*small*) | 0.99 [0.98 to 0.99] (*nearly perfect*) |
| | MuscleLab force platform | 268.8 ± 287.4 N | 268.4 ± 289.2 N | 261.5 ± 265.1 N | 0.12 [0.11 to 0.15] (*small*) | 0.99 [0.98 to 0.99] (*nearly perfect*) |
| $F_{peak}$ (n = 52) | Suiff sensor | 434.4 ± 282.4 N | 426.2 ± 277.2 N | 423.8 ± 252.2 N | 0.15 [0.13 to 0.18] (*small*) | 0.98 [0.97 to 0.99] (*nearly perfect*) |
| | MuscleLab force platform | 292.4 ± 292.4 N | 437.3 ± 291.1 N | 431.7 ± 265.4 N | 0.14 [0.12 to 0.17] (*small*) | 0.98 [0.97 to 0.99] (*nearly perfect*) |

Set data is presented as mean values (± standard deviation (SD)). Standardized typical error (TE) and ICC are displayed with 95% confidence intervals. $F_{mean}$ = Mean force during sets. $F_{peak}$ = Peak force during sets.

**Table 2. Comparison of $F_{mean}$ and $F_{peak}$ between MuscleLab 6000 force platform and Suiff pro sensor at isometric, 60BPM, 90BPM, 120 BPM conditions and the total pooled data.**

| Condition | Variable | MuscleLab force platform | Suiff sensor | Bias | TE | Correlation | Sig. |
|---|---|---|---|---|---|---|---|
| Isometric (n = 13) | $F_{mean}$ | 710.0 ± 258.0 N | 710.5 ± 263.7 N | 0.00 [-0.02 to 0.02] (*trivial*) | 0.03 [0.02 to 0.06] (*trivial*) | 1.00 [1.00 to 1.00] (*nearly perfect*) | < 0.001 |
| | $F_{peak}$ | 860.5 ± 299.1 N | 836.7 ± 296.6 N | -0.08 [-0.10 to -0.06] (*trivial*) | 0.03 [0.02 to 0.06] (*trivial*) | 1.00 [1.00 to 1.00] (*nearly perfect*) | < 0.001 |
| 60BPM (n = 13) | $F_{mean}$ | 111.5 ± 27.8 N | 114.3 ± 27.4 N | 0.10 [-0.06 to 0.25] (*trivial*) | 0.26 [0.14 to 0.51] (*small*) | 0.97 [0.89 to 0.99] (*nearly perfect*) | < 0.001 |
| | $F_{peak}$ | 287.5 ± 68.8 N | 283.1 ± 68.5 N | -0.06 [-0.18 to 0.05] (*trivial*) | 0.20 [0.11 to 0.38] (*small*) | 0.98 [0.94 to 0.99] (*nearly perfect*) | < 0.001 |
| 90BPM (n = 13) | $F_{mean}$ | 126.5 ± 34.0 N | 126.3 ± 33.9 N | -0.01 [-0.14 to 0.13] (*trivial*) | 0.23 [0.12 to 0.45] (*small*) | 0.97 [0.91 to 0.99] (*nearly perfect*) | < 0.001 |
| | $F_{peak}$ | 305.9 ± 86.5 N | 309.7 ± 79.8 N | 0.04 [-0.13 to 0.22] (*trivial*) | 0.29 [0.16 to 0.58] (*small*) | 0.96 [0.87 to 0.99] (*nearly perfect*) | < 0.001 |
| 120BPM (n = 13) | $F_{mean}$ | 127.2 ± 35.3 N | 126.9 ± 36.8 N | -0.01 [-0.10 to 0.09] (*trivial*) | 0.14 [0.08 to 0.27] (*small*) | 0.99 [0.96 to 1.00] (*nearly perfect*) | < 0.001 |
| | $F_{peak}$ | 314.7 ± 81.7 N | 310.6 ± 72.6 N | -0.05 [-0.25 to 0.15] (*trivial*) | 0.37 [0.20 to 0.72] (*moderate*) | 0.94 [0.81 to 0.98] (*nearly perfect*) | < 0.001 |
| ALL (n = 52) | $F_{mean}$ | 268.8 ± 287.4 N | 269.5 ± 288.5 N | 0.00 [0.00 to 0.01] (*trivial*) | 0.03 [0.02 to 0.03] (*trivial*) | 0.97 [0.94 to 0.98]* (*nearly perfect*) | < 0.001 |
| | $F_{peak}$ | 440.7 ± 292.4 N | 434.4 ± 282.4 N | -0.02 [-0.04 to 0.00] (*trivial*) | 0.07 [0.05 to 0.09] (*trivial*) | 0.96 [0.92 to 0.97]* (*nearly perfect*) | < 0.001 |

Data is presented as mean values (± standard deviation (SD)) and standardized mean bias, standardized typical error (TE) and correlation coefficient are displayed with 95% confidence intervals. Pearson correlations were performed when applicable. For non-parametrical data Spearman's rho was performed (indicated with an asterisk *). $F_{mean}$ = Mean force during sets. $F_{peak}$ = Peak force during sets.

The regression equation of $F_{peak}$ is:

$$Y = -7.8964 + (1.0328 \cdot X)$$

Fig 3B and 3D depict Bland-Altman plots illustrating the discrepancies between the two measurement systems concerning the assessment of $F_{mean}$ and $F_{peak}$, respectively.

## Discussion

The findings of this study show an excellent level of agreement between a versatile and affordable wireless force sensor (Suiff Pro) and a criterion measure (MuscleLab 6000 force platform) for the assessment of mean and peak forces during both isometric and dynamic RT at different paces.

In relation to the reliability of the measurement devices, a nearly perfect relationship was observed between the three sets in both both $F_{mean}$ and $F_{peak}$ measurements (see Table 1). A small TE was noted in all the variables measured with both devices. A possible factor contributing to TE in both devices is the inherent variability of human movement. In our study design, participants performed exercise sets three times under each condition, and although efforts were made to control the conditions to ensure consistency, human movement inherently exhibits slight variability from one repetition to another [27]. Consequently, this variation contributes to the observed TE when comparing two sets of exercises. Despite this inherent variability, the Suiff Pro sensor demonstrated excellent reliability for both $F_{mean}$ and $F_{peak}$, akin to the reliability exhibited by the MuscleLab force platform.

Overall, the Suiff Pro force sensor demonstrated nearly perfect relationships with the pooled data gathered with the force platform along with trivial bias and TE, both for mean and peak force values (see Table 2). Although the bias was trivial and the level of agreement was very high in all of the analysed conditions, there were certain differences in the level of agreement between parameters (i.e., peak or mean force values), and also an effect of the pace at which the exercise was executed. In isometric conditions, the most controlled set-up, the agreement between systems was nearly perfect both for mean and peak forces (see Table 2). Moreover, in dynamic conditions, the TE for $F_{peak}$ increased at higher paces (60BPM < 90BPM < 120 BPM) and their values tended to be higher compared to the $F_{mean}$ values (see Table 2). A plausible explanation for the discrepancies is the difference in sampling rates between sensors (Suiff Pro works at 100 Hz and MuscleLab 6000 at 1000 Hz). Previous research has shown that this parameter affects the level of agreement between measurement systems [28].

Our results indicate that bias was trivial in all conditions, implying that the divergence from the actual force values would be negligible while assessing a large group of athletes, a given population, or multiple samples of the same athlete.

TE was observed to be higher for peak forces assessed at faster execution paces. Thus, researchers and practitioners should be mindful of this in cases of low number of measurements or faster exercise pace. Consequently, to reduce the potential measurement errors, it is advisable to take repeated measurements (e.g., two or three trials) when assessing peak forces in fast movements for individual athletes.

This study is not without limitations. We are aware that the sample size is small. However, it must be noted that the statistical power for correlation analysis was above the recommended (>0.8). It would be interesting for future studies to test the force sensor in different populations and with other resistance exercise devices.

In conclusion, the Suiff Pro force sensor was found to be a valid instrument for assessing forces in both isometric and dynamic tasks, providing practitioners with a reliable tool for monitoring kinematic variables of RT exercises.

## Conclusions

Practitioners must be aware of strengths and limitations of devices utilised for the assessment of force outputs prior to their use during sports training and rehabilitation interventions. Despite the existence of other valid methods to assess force outputs during RT exercises [8, 29, 30], Suiff Pro sensors have present advantages (e.g., affordable, practical, portable and versatile) that might be difficult to find in alternative devices. Additionally, this sensor has the capability to provide live augmented feedback which may increase motivation during RT, in turn enhancing the subsequent physical adaptations [4]. Therefore, this sensor may be considered as a good option for practitioners for daily monitoring of force outputs during RT exercises.

## Practical applications

The validation of the wireless force sensor presented in this study under isometric and dynamic conditions expands the possibilities for the assessment and control of training loads using elastic bands, an aspect that has been less extensively quantified and studied thus far. Furthermore, as a prospective avenue, it opens the possibility to investigate the validity of these strength assessment systems in other training modalities such as flywheels, pneumatic systems, suspension training, etc.

## Supporting information

**S1 Data.**
(XLSX)

## Acknowledgments

The authors would like to thank the volunteers who participated in the study.

## Author Contributions

**Conceptualization:** Víctor Illera-Domínguez, Lluís Albesa-Albiol, Jorge Castizo-Olier, Adrián Garcia-Fresneda, Bernat Buscà, Carlos Ramirez-Lopez, Bruno Fernández-Valdés.

**Data curation:** Víctor Illera-Domínguez, Lluís Albesa-Albiol, Jorge Castizo-Olier, Adrián Garcia-Fresneda, Bernat Buscà, Carlos Ramirez-Lopez, Bruno Fernández-Valdés.

**Formal analysis:** Víctor Illera-Domínguez, Lluís Albesa-Albiol, Jorge Castizo-Olier, Adrián Garcia-Fresneda, Bernat Buscà, Carlos Ramirez-Lopez, Bruno Fernández-Valdés.

**Investigation:** Víctor Illera-Domínguez, Lluís Albesa-Albiol, Jorge Castizo-Olier, Adrián Garcia-Fresneda, Bernat Buscà, Carlos Ramirez-Lopez, Bruno Fernández-Valdés.

**Methodology:** Víctor Illera-Domínguez, Lluís Albesa-Albiol, Jorge Castizo-Olier, Adrián Garcia-Fresneda, Bernat Buscà, Carlos Ramirez-Lopez, Bruno Fernández-Valdés.

**Project administration:** Víctor Illera-Domínguez, Lluís Albesa-Albiol, Jorge Castizo-Olier, Adrián Garcia-Fresneda, Bernat Buscà, Carlos Ramirez-Lopez, Bruno Fernández-Valdés.

**Resources:** Víctor Illera-Domínguez, Lluís Albesa-Albiol, Jorge Castizo-Olier, Adrián Garcia-Fresneda, Bernat Buscà, Carlos Ramirez-Lopez, Bruno Fernández-Valdés.

**Software:** Víctor Illera-Domínguez, Lluís Albesa-Albiol, Jorge Castizo-Olier, Adrián Garcia-Fresneda, Bernat Buscà, Carlos Ramirez-Lopez, Bruno Fernández-Valdés.

**Supervision:** Víctor Illera-Domínguez, Lluís Albesa-Albiol, Jorge Castizo-Olier, Adrián Garcia-Fresneda, Bernat Buscà, Carlos Ramirez-Lopez, Bruno Fernández-Valdés.

**Validation:** Víctor Illera-Domínguez, Lluís Albesa-Albiol, Jorge Castizo-Olier, Adrián Garcia-Fresneda, Bernat Buscà, Carlos Ramirez-Lopez, Bruno Fernández-Valdés.

**Visualization:** Víctor Illera-Domínguez, Lluís Albesa-Albiol, Jorge Castizo-Olier, Adrián Garcia-Fresneda, Bernat Buscà, Carlos Ramirez-Lopez, Bruno Fernández-Valdés.

**Writing – original draft:** Víctor Illera-Domínguez, Lluís Albesa-Albiol, Jorge Castizo-Olier, Adrián Garcia-Fresneda, Bernat Buscà, Carlos Ramirez-Lopez, Bruno Fernández-Valdés.

**Writing – review & editing:** Víctor Illera-Domínguez, Lluís Albesa-Albiol, Jorge Castizo-Olier, Adrián Garcia-Fresneda, Bernat Buscà, Carlos Ramirez-Lopez, Bruno Fernández-Valdés.

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
