## [Decision Letter · Decision Letter 0]

22 Nov 2023

PONE-D-23-31360Reliability and validity of a low-cost, wireless sensor and smartphone app for measuring force during isometric and dynamic resistance exercises.PLOS ONE

Dear Dr. Albesa,

Thank you for submitting your manuscript to PLOS ONE. After careful consideration, we feel that it has merit but does not fully meet PLOS ONE’s publication criteria as it currently stands. Therefore, we invite you to submit a revised version of the manuscript that addresses the points raised during the review process.

We look forward to receiving your revised manuscript.

Kind regards,

Julio Alejandro Henriques Castro da Costa

Academic Editor

PLOS ONE

2. Please include the reference section of your manuscript. 

3. We note that Figure(s) 1 and 2 in your submission contain copyrighted images. All PLOS content is published under the Creative Commons Attribution License (CC BY 4.0), which means that the manuscript, images, and Supporting Information files will be freely available online, and any third party is permitted to access, download, copy, distribute, and use these materials in any way, even commercially, with proper attribution. For more information, see our copyright guidelines: http://journals.plos.org/plosone/s/licenses-and-copyright.

a. You may seek permission from the original copyright holder of Figure(s) 1 and 2 to publish the content specifically under the CC BY 4.0 license. 

Reviewers' comments:

Reviewer's Responses to Questions

**Comments to the Author**

1. Is the manuscript technically sound, and do the data support the conclusions?

Reviewer #1: Yes

Reviewer #2: Yes

2. Has the statistical analysis been performed appropriately and rigorously? 

Reviewer #1: Yes

Reviewer #2: Yes

3. Have the authors made all data underlying the findings in their manuscript fully available?

Reviewer #1: No

Reviewer #2: Yes

4. Is the manuscript presented in an intelligible fashion and written in standard English?

Reviewer #1: Yes

Reviewer #2: Yes

5. Review Comments to the Author

Reviewer #1: General:

The manuscript is extremely well written, with zero glaring English or formatting issues.

There are no major issues with this manuscript as each section does a great job and is of high quality. For this is commend the authors, and can only come up with minor revisions!

However, I must also include that there are several other modern strain-gauges (or similar) that have recently been examined for validity and reliability. Therefore, this study is not particularly special. Although I do recognize the unique way that the researchers examined dynamic force output through the use of elastic bands!

Title:

The title is good, no edits needed.

Abstract:

Not a big issue, but not sure why the abstract is broken into separate paragraphs.

Line 35: I’m not sure if bias can actually be zero. Might this be better expressed as “<0.01” or similar?

The abstract should be clearer regarding the isometric and dynamic ‘vertical row’. Would this be more accurate to say ‘upright row’?

Introduction:

Beautifully written. I have no revisions required in this section!

Methods:

Line 102: the authors write ‘upright rows’. This is great, and far better than ‘vertical row’ that is in the abstract.

The sample size of 13 is relatively small, especially for a fairly simplistic study. It is not a huge flaw, however, if the paper is not accepted, I would recommend the authors increase their sample to ~20 (unless they can present a compelling argument for why 13 is sufficient.

Line 117: I am not sure what the authors mean by “…resisted by a slack at…”

The use of elastic bands is very cool!

I really like the choices of statistical analysis! I am almost certain the figures were not made in JASP. Thus the authors should state how the figures were made.

Results:

No issues with the results. The figures and tables are good.

Discussion:

The discussion is also mostly on-point, well done.

However, I believe that the use of elastic band to assess dynamic force is VERY interesting. The authors should elaborate on this potential practical application of the device, and point out that other similar devices have NOT been validated in this way (typically only isometric outputs).

Reviewer #2: The research is very interesting and useful for all people using this type of devices. According to the Reviewer, in future similar studies, the number of the study group should be increased. A control group consisting of, for example, non-exercising people could also be included. To check whether these measurements are equally accurate in both groups.

6. PLOS authors have the option to publish the peer review history of their article (what does this mean?). If published, this will include your full peer review and any attached files.

Reviewer #1: **Yes: **Dustin J Oranchuk

Reviewer #2: No

---

## [Author Response · Author response to Decision Letter 0]

24 Jan 2024

Dear reviewers,

We extend our sincere gratitude for your diligent efforts and valuable time invested in reviewing our manuscript. Your insightful recommendations have been thoughtfully incorporated, significantly enhancing the overall quality of the document.

For your convenience, we have provided a detailed response in the attached files, addressing each of the specific comments made.

Thank you once again for your valuable contributions.

Best regards,

Lluis Albesa

---

## [Decision Letter · Decision Letter 1]

1 Feb 2024

Reliability and validity of a low-cost, wireless sensor and smartphone app for measuring force during isometric and dynamic resistance exercises.

PONE-D-23-31360R1

Dear Dr. Albesa,

We’re pleased to inform you that your manuscript has been judged scientifically suitable for publication and will be formally accepted for publication once it meets all outstanding technical requirements.

Kind regards,

Julio Alejandro Henriques Castro da Costa

Academic Editor

PLOS ONE

Additional Editor Comments (optional):

Reviewers' comments:

Reviewer's Responses to Questions

**Comments to the Author**

1. If the authors have adequately addressed your comments raised in a previous round of review and you feel that this manuscript is now acceptable for publication, you may indicate that here to bypass the “Comments to the Author” section, enter your conflict of interest statement in the “Confidential to Editor” section, and submit your "Accept" recommendation.

Reviewer #1: All comments have been addressed

2. Is the manuscript technically sound, and do the data support the conclusions?

Reviewer #1: Yes

3. Has the statistical analysis been performed appropriately and rigorously? 

Reviewer #1: Yes

4. Have the authors made all data underlying the findings in their manuscript fully available?

Reviewer #1: Yes

5. Is the manuscript presented in an intelligible fashion and written in standard English?

Reviewer #1: Yes

6. Review Comments to the Author

Reviewer #1: I thank the authors for completing the minor revisions to a satisfactory level. I have no further comments or revisions that require addressing.

Congratulations.

7. PLOS authors have the option to publish the peer review history of their article (what does this mean?). If published, this will include your full peer review and any attached files.

Reviewer #1: **Yes: **Dustin J Oranchuk

---

## [Editor Report · Acceptance letter]

11 Mar 2024

PONE-D-23-31360R1 

PLOS ONE

Dear Dr. Albesa-Albiol, 

I'm pleased to inform you that your manuscript has been deemed suitable for publication in PLOS ONE. Congratulations! Your manuscript is now being handed over to our production team.

Kind regards, 

on behalf of

Dr. Julio Alejandro Henriques Castro da Costa 

Academic Editor

PLOS ONE